# School Climate and Peer Victimization. Involvement, Affiliation and Help Perceived in School Centers as Protective Factors against Violent Behavior in Adolescent Couples

**Marta Ruiz-Narezo ***  and **Rosa Santibáñez Gruber ***

Department of Social Pedagogy and Diversity, University of Deusto, 48007 Bilbao, Spain

\* Correspondence: marta.ruiznarezo@deusto.es (M.R.-N.); rosa.santibanez@deusto.es (R.S.G.);
  Tel.: +34-944-139-000 (ext. 2121) (M.R.-N.); +34-944-139-000 (ext. 2280) (R.S.G.)

**Abstract:** This article presents the results of a non-experimental, quantitative cross-sectional study conducted on an adolescent group. The sample of adolescents was acquired from high schools and vocational training, where the relationship between the school climate, more specifically, the involvement, affiliation, and perception of help and violence that is both experienced and exercised between partners. The study sample consisted of 433 adolescents aged 12–19 years from four educational centers from a municipality of Greater Bilbao. Since there are analyses that refer specifically to romantic relationships, in those cases, the 67.7% (N = 275) of the sample that claims to have or have had a romantic relationship is considered. Finally, there was evidence to suggest the existence of influence between the school climate and the implication of violence in adolescent couples.

**Keywords:** classroom behavior/environment; school context; peer relationships; dating/dating violence; violence/violent behaviors

## 1. Introduction

Peer violence has clear negative consequences for everyone involved, whether they are minors or adults. Context is an important influence as a risk or protective factor for early adolescents to see or not see themselves as engaging in violent behavior of various kinds [1]. The aim of this article is to examine, through a quantitative cross-sectional study, the phenomenon of the influence exerted by a positive school climate by analyzing the affiliation, help, and involvement perceived in the school center on the part of peers and teachers, and the relationship between this and violent behavior in adolescent couples, both from the role of victim and aggressor.

This research starts in early adolescence and is projected throughout adolescence, that is, during middle and late adolescence. The period of adolescence is considered one that is associated with risks and problems. Adolescence is defined [2] as the process of becoming older or achieving autonomy, responsibility, and psychological and social adulthood; it is a complex period in which difficulties occur at different levels: individual, social-community, school, peer group, and family. Adolescence is, therefore, a complex stage in which minors are influenced by multiple circumstances and by different people and contexts, while simultaneously being involved in the process of maturing, learning, and developing their personal identities. All of this together makes the family and school center vital [3,4].These authors [5] also considered the perception of adolescent invulnerability as relevant, which contributes to greater engagement in risk behavior. It therefore appears that the current image of adolescence is framed around numerous problems that go beyond the limits of social control.

Risk behavior is considered as a manifestation of underlying social problems in which different factors or agents that condition the lives of adolescents converge [6,7]. They do not emerge by chance. Rather, they appear because of a series of personal and contextual circumstances that directly affect adolescents [8]. There is, for example, a clear continuity and relationship between the patterns of external behavior in childhood and the acquisition of risk in early adolescence [9].

It is necessary to indicate that prompt educational intervention, centered on early adolescence, will help ensure that problems which arise do not trigger more serious issues and remain as mere experimentation processes. Thus, for example, victimization between peers, a form of stress, harassment, and aggression will always have negative impacts for the minors involved, but if we manage to intervene preventively in early adolescence, it will be possible to avoid even more negative experiences and consequences, as well as reeducate minors on patterns of respect and coexistence and avoid, a posteriori, having them become violent adults in relationships with their romantic partners in particular, as well as other general relationships.

In this article, we will refer to one of the violent behaviors among peers that, at this moment, is creating a high level of social alarm: violent behavior in adolescent couples ([10] p. 8):

> *Gender violence is a problem of deep social significance. The topic's importance is generating diverse studies, legislative and judicial measures, social protective mechanisms, educational, and therapeutic strategies for victims, as well as specific interventions with the aggressors. In the educational field there is widespread concern about the relationship that this problem may imply for boys and girls, and also the role that schools and social education may have in preventing it.*

When we refer to violence in couples, we refer to any type of violence committed by any sex against the other person of the same or different sex, but with a different manifestation of gender [11]. According to experts, the most common types of violence in couples are physical, psychological, sexual and economic [12,13]. Violence in adolescent couples differs from violence in adult relationships. Although data on the victimization of women are overwhelming in adult relationships, in adolescence, the data show important nuances [14,15] as both victims and aggressors are more balanced among boys and girls and primarily because psychological violence surfaces, which can lead to sexual and physical violence, with economic violence being very unusual. As indicated in ([10] p. 83):

> *If we look at self-registration studies among adolescents and youths, the results increase significantly in terms of incidence percentages. Around 90% of romantic relationships have verbal aggression and physical aggression occurs in 40% (Povedano, 2013). Studies conducted on violence in secondary schools obtain similar results in terms of the incidence of these behaviors: verbal aggression and social exclusion are the most frequent types of violence (Ararteko, 2006; State Observatory of School Coexistence, 2010). In this same line of psychological violence, we find that most recent research on romantic relational aggression, which is defined as acts carried out with the pretense of ignoring, excluding, preventing participation, or spreading rumors in an attempt to damage the self-esteem of an adversary, friendships or social status (Shaffer, 2002). Linder, Crick, and Collins (2002) found no differences in this type of violence according to gender, obtaining similar results in subsequent studies (Bagner, Storch, & Preston, 2007, Kuppens, Grietens, Onghen, Michiels, & Subramanian, 2008).*

From this point of view, considering the relevance and social significance as well as future repercussions that chronicity and acceptance of violence can have as a regular core concept in relationships, it seems necessary to analyze how and from what context we can intervene to stop this. This study [16] among others, has been used as references. It seems that the problem arises from the adolescents themselves and that they are the ones who support or impede their own development. However, it is known that in the personal and social adjustment of youth, adults, and, especially, the family and school play a fundamental role, and, therefore, it is not possible to claim that only the adolescent is responsible [17]. Therefore, we will now delve into school influence, given the importance that this learning space can have in the lives of adolescents, as a protective factor against engaging in violent behavior in romantic relationships, either in the role of victim or the aggressor [18,19];

## 2. Educational Centers as a Place of Protection against Violent Behavior in Romantic Relationships

Educational centers are considered an important developmental context for adolescents [20]. Adolescents go to school with a large repertoire of beliefs, values, and internalized behavior (both positive and negative) derived from the adolescents themselves (personal factors) and their families (family factors). When they reach these educational centers, they experience the importance of peer groups [21] in addition to the importance and impact of educational centers in their lives [22] and in the development patterns of healthy behavior, as a preventive context [23] since it is the place where, besides basic academic content, values, attitudes [24,25] and learnings complementary to those introduced by the family unit are transmitted [17]. In addition, adolescents' interaction with other adults and their peers with whom they form important bonds of friendship occurs in educational centers [26,27].These centers are not in themselves factors or agents that influence the development of a certain risk behavior, but are an environment, safe in principle, for the development in which various situations that may support or reduce the appearance of problem behavior occur, therefore working as a risk or protective factor.

According to Musitu et al. [4], schools have to achieve five priority objectives: transform adolescents into reflective people, prepare them for the commitment with the working world, make them citizens who aptly fulfill their duties, train them on an ethical level, and to thus make them physically and psychologically healthy people. Therefore, following this premise, it could be said that when formal education does not fulfill its function of socialization and development and fails to promote the learning of competences, values, attitudes, and the capacity for critical reflection, adolescents will be at greater risk of being absorbed in problem behaviors. In the same way, the behavior of all people involved in the school (teachers, principal, cafeteria workers, cleaners, janitors, etc.) can play the role of predictor, moderator, or instigator in relation to variables of risk behavior or appropriate and healthy behavior for adolescents. It is therefore understood that educational centers [28], are reproductive and transformative. Also, as indicated by Vera [29], the functions of schools have simultaneously grown and evolved and, as a direct consequence of the delimitation of the functions of families and communities, have gradually acquired a greater role in adolescent development.

Schools, therefore, can be understood as places in which to develop protective variables, favoring social inclusion through educational support [26] the appropriate school climate, and a positive position. A correct pedagogical methodology based on inclusion, equality, and diversity that understands the difficulties of each student and tries to help the student overcome them from a positive perspective, relying on all the agents that participate in the school and always considering the family, will support the decrease of risk behavior as indicated by some authors [26,30–32]. Likewise, the importance of structural and organizational characteristics must be noted, as well as the teaching of a democratic education [24,32]. The organizational structure of centers and the importance they assign to the relationships they create, as well as the rules of coexistence, influence both the students and the rest of the educational community, family, and society. Thus, a positive school climate, with an adequate degree of cohesion, will support social integration, help, and mutual support, thus appreciating teaching-learning and collaboration [24,32,33]. Educational centers offer instruction not only in academic matters, but also in attitudes and values, and it is necessary to develop them adequately and transversally, without detracting from academic subjects [32].

We must mention the importance taken on by peer groups during adolescence as one of the most important social contexts [16]. If these peer relations maintain positive interactions, they significantly reduce the chances of committing risk behaviors by exercising positive behavioral modeling [34]. If the peer group, usually located in the school center, commits violent acts or transgresses norms, adolescents will be expected to do the same, with the objective of belonging to the group. Fiske [35] established that groups not accepted or understood as normalized often tend to show, in addition to negative attributions, stereotypes that in turn influence greater participation in negative affective behaviors and responses due to the need to continue belonging to the group. In contrast, if positive interactions, high

degree of social support, and good interpersonal relationships are maintained with other partners, they will noticeably reduce the chances of committing problem behaviors [24,33,36–38].In addition, it will be important to work on the rules of coexistence among the students themselves with the intention that they respect diversity and, thus, behavior based on violence does not appear [30]. Table 1 summarizes the main school protective factors:

**Table 1.** School Protective Factors.

| PROTECTIVE FACTORS | AUTHORS |
|---|---|
| School climate and positive position toward school, with feeling of belonging and unity | [24,32,33] |
| Social inclusion | [16,26] |
| Educational support | [16,26] |
| School participation (friendship with classmates, involvement in the classroom, etc.) | [17,30,31] |
| Social support | [37] |
| Performing tutorials, meetings with the school community, changing disciplinary procedures, successful school performance, and recognizing achievements and social norms that discourage violence and drug use | [24,33,38–40] |
| Involvement in volunteering | [30] |
| Relationship with peer groups with positive and adequate behavior and respect for norms | [16,33,38] |
| Feeling of belonging to a prosocial group | [24] |
| Good interpersonal relationships | [36] |

The educational center is a privileged setting where students can observe conflicts between students, situations of isolation or social rejection [41,42], as well as risk behaviors that may be taking place. These conflicts must be attended to by the school center, including the rest of the agents involved [40]. Thus, school mediation programs, which involve the peer group, aim to give adolescents a voice and work with them in various competences [43] to conduct preventive interventions based on collective support, cooperative models, and zero tolerance in situations of violence or social exclusion [30,38]. It has been proven that poor academic performance, together with other variables such as a bad school climate, a bad attitude toward the future, and education and engagement in acts of rebellion will support the emergence of risk behavior [44].

Educational centers are one of the places where minors spend most of their time. They should therefore be understood as spaces that generate learning, which, through daily practice, encourage adolescents to learn conflict resolution, development of social skills, as well as how to expand their cognitive and relational capacities and build citizenship, acquiring capacities to generate alternatives to the actions that arise from the commitment to the environment and the global community [45]. Therefore, risk behavior or violent behavior endangers stability, balance, and, consequently, the proper development of adolescents at the biopsychosocial level [46].

Given the importance that violence in couples' relationships is gaining in our context among the adolescent population, and taking into account that these adolescents spend a large part of their time at school, we wondered if a good school climate could have a positive influence on a lesser involvement in violent behavior in couples' relationships. We wanted to look in depth at the possibility that there were differences between boys and girls, and differences according to the educational itinerary they belonged to. We analyzed the population of secondary education and vocational training in a municipality of Greater Bilbao.

## 3. Methodology

### 3.1. Design

This is a non-experimental, cross-sectional research approach that uses quantitative methodology. The instrument used is an anonymous and self-recorded questionnaire, consisting of 40 closed questions that are applied to a sample of 433 students (199 girls, 213 boys, 21 unspecified) from ESO (Educación Secundaria Obligatoria-Compulsory Secondary Education) and FPB (Formación Profesional Básica-Vocational Training Education) between the ages of 12 and 19 in four educational centers of a municipality in Greater Bilbao. Since there are analyses that refer specifically to romantic relationships, in those cases, the 67.7% (N = 275) of the sample that claims to have or have had a romantic relationship is considered.

### 3.2. Participants

We started with a sample of 433 people and only included students who indicated having or having had a partner. Specifically, we referred to 275 cases, that is, 67.7% of the total sample. Regarding gender, of the 275 students, 130 were men and 137 women, and eight people did not indicate their sex. In relation to the educational track, 60% (48 people) of the FPB students were men (50% native and 50% of immigrant origin) while 40% (32) were women. Among these 32 FPB women, 37.5% were of immigrant origin, compared to 62.5% of the natives. In ESO, 43.9% (82) were men, with 17% of students being from immigrant origin and 83% being natives, and 56.1% (105) were women, with a high percentage of natives (89.5%) compared to immigrant origin (10.5%). Regarding the age, it should be noted that in the sample of 275 students, 44% (121 students) were categorized as being in early adolescence as they were between 12 and 14 years old when the research was conducted. Moreover, we recorded data on older students in middle and late adolescence. In particular, the following data were recorded related to middle adolescence: 131 students, corresponding to 47.6%. Referring to late adolescence or youth, the sample was reduced to a small group (23 students or 8.36%) who were between 18 and 20 years old.

### 3.3. Variables and Instruments

The questionnaire consists of 40 questions that form 288 items. We have conducted an in-depth analysis of the Spanish adaptation [47] of the Classroom Environment Scale (CES), Social Climate Scales [48], specifically for the "Relationships" subscale, which consists of the following dimensions: affiliation, involvement, and help. Affiliation refers to the level of friendship between students and the extent to which they help each other in their tasks, how well they know one another, and enjoy working together. Involvement measures the degree to which students express interest in class activities, participate in the classroom, and enjoy the environment created. Finally, help refers to how much assistance, concern, and friendship teaching staff offers to students (open, fluid communication, with confidence, and the opportunity for free expression).

Likewise, we have also analyzed the scale related to measuring violent behavior in romantic relationships between adolescents from a dual role: victim and aggressor. The two versions of the question (role of victim, role of aggressor) propose up to six situations in the case of violent behavior in romantic relationships that are ordered gradually from insults, online aggression, up to physical aggression. Cronbach's Alpha reliability coefficients obtained in the scales are adequate, ranging between 0.691 and 0.783 (Table 2).

**Table 2.** Description of the instruments.

| Scale | Dimension [1] (Elements); $\alpha$ |
|---|---|
| **CES**. (The Social Climate Scales: Classroom). "Relationships" Scale Spanish adaptation [47] based on the original scale [48] | Affiliation (10 items): 0.693 Help (10 items): 0.691 Involvement (10 items): 0.693 |
| **VIOLENCE IN THE COUPLE** [15] | Violence, victim role (7 items): 0.763 |
| | Violence, aggressor role (7 items): 0.780 |

[1] We attach in an annex (Supplementary Materials) the detailed information of the items consulted in this investigation, bassed on **CES** (The Social Climate Scales: Classroom) -"Relationships" Scale Spanish adaptation [47] based on the original scale [48] and violence in the couples [15] We attach the original items used in Spanish.

*3.4. Ethical Issues*

The study has been carried out observing the regulations in force and the principles of ethics in all matters relating to the protection and avoidance of risks to participants and respect for autonomy. Furthermore, the methodological, ethical and legal principles that are specific to and obligatory for this type of research were taken into account. In each of the measurements carried out, once the educational centers had been selected, the management teams and the families were informed of the objectives and were asked for informed consent, with information about the research, an e-mail and a telephone number for any doubts. Once consent was obtained, the application of the instruments, conducted collectively in the classrooms, insisted on the confidentiality and anonymity of the responses. The sessions lasted an average of 55 min and were held without the presence of the usual teaching staff. The overall descriptive results were returned to the AMPAS (parents' associations) and/or the Management Teams of the four participating centers.

*3.5. Procedure*

In relation to the procedure, the study has been developed respecting current regulations and the principles of ethics in everything related to the protection and avoidance of risks to participants and respect for autonomy. Likewise, the appropriate and obligatory methodological, ethical, and legal principles in this type of research were considered. In this and each of the measurements performed, once the educational centers were selected, the research objectives were reported to the administrative teams and families from whom informed consent was obtained from all individual participants included in the study. This was acquired through a document that contained information about the research, its objectives, an email address, and a contact phone number to answer any questions. Once this was obtained, in the session to apply the instruments, carried out collectively in the classrooms, the confidentiality and anonymity of the answers was emphasized. The sessions lasted an average of 55 min and were carried out without the presence of the usual teaching staff. The overall descriptive results were returned to the AMPAS and/or the Administrative Teams once the results were obtained.

**4. Analysis and Results**

Next, the results of the Pearson $\chi^2$ statistical calculations are shown, as well as Matthews's Phi correlation coefficient (Indicator of the direction of the relationship) and Pawlik's corrected contingency coefficients (Shows the relative amount of the association $Cc = \frac{C}{Cmax}$ $C = \sqrt{\frac{X^2}{n+X^2}}$ $Cmax = \sqrt{\frac{k-1}{k}}$) (Cc). The results obtained in the association analysis developed for the three school variables—affiliation, help, and involvement—and the measurement items used in the scales of violence in adolescent romantic relationships, both from the role of victim and the aggressor, are presented. All the data will be presented according to gender and educational track, referring first to the results obtained for the girls and then for the boys of both educational tracks.

*4.1. Affiliation with the School and Relationships Observed with Violence in an Adolescent Couple (Victim and Aggressor)*

Affiliation (CES) is described as the level of friendship between students and how they help each other in their tasks, how well they know each other and enjoy working together.

Female student body. For FPB girls, the results show a significant ($\chi^2$ = 5.448, $p$ = 0.020, Cc = 0.545) and inverse ($\Phi$ = −0.419) relationship between the degree of affiliation in the school and having felt themselves to be a victim of control and isolation by their partner or ex-partner. Likewise, a significant ($\chi^2$ = 5.420, $p$ = 0.020, Cc = 0.537) and inverse ($\Phi$ = −0.412) relationship is observed between the degree of affiliation and having felt themselves to be a victim of violence by their partner or ex-partner, referring to the scale that includes all the items analyzed. Finally, we note the significant ($\chi^2$ = 5.387, $p$ = 0.020, Cc = 0.543) and inverse ($\Phi$ = −0.417) relationship between the degree of affiliation in the school and being recognized as an aggressor of controlling behavior and isolation of friendships against their partner or ex-partner.

By observing ESO girls, similarities in data are found for items related to the area of affiliation in the school, which is similar to female FPB students. A significant ($\chi^2$ = 10.703, $p$ = 0.001, Cc = 0.445) and inverse ($\Phi$ = −0.332) relationship is observed between the degree of affiliation in the school and having felt themselves to be a victim of control and isolation by their partner or ex-partner. Significant ($\chi^2$ = 4.273, $p$ = 0.039, Cc = 0.289) and inverse ($\Phi$ = −0.210) relationships are observed between affiliation and victimization in behavior where ESO girls have felt themselves to be victims of humiliation, insults, and threats through social networks or harassment over the phone, in addition to significant ($\chi^2$ = 4.273, $p$ = 0.039, Cc = 0.289) and inverse ($\Phi$ = −0.210) relationships between affiliation and victimization, feeling obliged to do things they did not want to do. In the same manner, similarities are observed again in the relationships between ESO and FPB girls when referring to the significant ($\chi^2$ = 5.332, $p$ = 0.020, Cc = 0.265) and inverse ($\Phi$ = −0.192) relationship between the degree of affiliation and having felt themselves to be a victim of violence by their partner or ex-partner and the scores in the total scale.

Male student body. FPB men show significant ($\chi^2$ = 7.822, $p$ = 0.005, Cc = 0.565) and inverse ($\Phi$ = −0.437) relationships between the degree of affiliation in the school and recognizing oneself as an aggressor, forcing their partner or ex-partner to do things they did not want to do. A significant ($\chi^2$ = 7.822, $p$ = 0.005, Cc = 0.584) and inverse ($\Phi$ = −0.437) relationship is also observed in relation to the affiliation and involvement of FPB boys as aggressors, forcing their partner or ex-partner to engage in sexual behaviors against their will.

Finally, we mention the ESO boys. Similar to the case with ESO girls, there is a greater number of significant relationships than with students in the FPB track. They show significant ($\chi^2$ = 11.970, $p$ = 0.001, Cc = 0.512) and inverse ($\Phi$ = −0.389) relationships between the degree of affiliation and feeling of humiliation, being insulted, or publicly threatened by social networks and feeling obliged to do things they did not want to do ($\chi^2$ = 11.970, $p$ = 0.001, Cc = 0.512, and inverse: $\Phi$ = −0.389). From the perspective of the role of aggressor, significant and inverse relationships are observed between affiliation in the school and the following behaviors: forcing their partner or ex-partner to do things that they did not want to do; humiliating, insulting or threatening publicly through social networks and/or harassing on the phone; compelling the person to perform sexual behaviors that they did not to engage in, all of which obtained a significant ($\chi^2$ = 11.970, $p$ = 0.001, Cc = 0.512) and inverse ($\Phi$ = −0.389) relationship, in addition to showing a significant ($\chi^2$ = 4.575, $p$ = 0.032, Cc = 0.329) and inverse ($\Phi$ = −0.241) relationship between affiliation in the school and being considered an aggressor of physical violence, acknowledging having beaten their partner or ex-partner. Below we present Table 3, which indicates the significant results obtained in the analysis of the affiliation in the school variable and violent behavior in romantic relationships (from the dual role: victim and/or aggressor), with the behaviors categorized as follows: psychological violence, physical violence, and sexual violence.

**Table 3.** Table of Significant Results in the Analysis of the Affiliation Variable (CES) and the Observed Relationship with Violence in Adolescent Couples.

| Violence in Adolescent Couples | | | CES Affiliation | | | |
|---|---|---|---|---|---|---|
| | | | FPB Woman | ESO Woman | FPB Man | ESO Man |
| Psychological violence | "He or she has tried to control me and isolate me from my friendships (forbidding me to see someone, etc.)" 34.A. B | $\chi^2$ (p-value) N Cc Φ (p-value) Φ | 5448 0.020 31 0.545 −0.419 0.020 | 10,703 0.001 97 0.445 −0.332 0.001 | - | - |
| Psychological violence | "I have been humiliated, insulted, or publicly threatened on social networks or harassed by phone" 34.A. D | $\chi^2$ (p-value) N Cc Φ (p-value) Φ | - | 4273 0.039 97 0.289 −0.210 0.039 | - | 11,970 0.001 79 0.512 −0.389 0.001 |
| Sexual violence | "I felt obligated to perform sexual acts that I did not want to do" 34.A. E | $\chi^2$ (p-value) N Cc Φ (p-value) Φ | - | 4273 0.039 97 0.0289 −0.210 0.039 | - | 11,970 0.001 79 0.512 −0.389 0.001 |
| SCALE: BEING A VICTIM OF VIOLENCE IN THE COUPLE | | $\chi^2$ (p-value) N Cc Φ (p-value) Φ | 5420 0.020 32 0.537 −0.412 0.020 | 5332 0.021 145 0.265 −0.192 0.021 | - | - |
| Psychological violence | "I have tried to control (monitoring his or her cell phone, etc.) and isolate my partner from friends" (forbidding him or her to see someone, etc.) 34.B. B | $\chi^2$ (p-value) N Cc Φ (p-value) Φ | 5387 0.020 31 0.543 −0.417 0.020 | - | - | - |
| Psychological violence | "I forced my partner to do things he or she did not want to do (change clothes, because those he or she was wearing did not seem good to me, etc.)" 34.B. C | $\chi^2$ (p-value) N Cc Φ (p-value) Φ | - | - | 7822 0.005 41 0.565 −0.437 0.005 | 11,970 0.001 79 0.512 −0.389 0.001 |
| Psychological violence | "I have humiliated, insulted, or threatened my partner publicly or harassed him or her by phone" 34.B. D | $\chi^2$ (p-value) N Cc Φ (p-value) Φ | - | - | - | 11,970 0.001 79 0.512 −0.389 0.001 |
| Sexual violence | "I forced my partner to perform sexual acts that he or she did not want to do" 34.B. E | $\chi^2$ (p-value) N Cc Φ (p-value) Φ | - | - | - | 11,970 0.001 79 0.512 −0.389 0.001 |
| Physical violence | "I have hit my partner" 34.B. E | $\chi^2$ (p-value) N Cc Φ (p-value) Φ | - | - | - | 4575 0.032 79 0.329 −0.241 0.032 |

More information about the questions used in Supplementary Materials.

### 4.2. Help Perceived in the School and Relationships Observed with Violence in Adolescent Couples (Victim and Victimizer)

Second, the results obtained in relation to help (CES) are shown. Help refers how much assistance, concern, and friendship of the teacher was perceived by students (open communication with students, trust in them, and an interest in their ideas, for example), (Table 4).

Female student body. In the case of FPB girls, the results show the existence of a significant ($\chi^2$ 4.118, $p = 0.042$, Cc = 0.497) and inverse ($\Phi = -0.377$) relationship between the degree of help at the school and perceiving oneself as an aggressor in adolescent romantic relationships. There are no significant relationships among students ESO girls.

Male student body. The FPB men do not show significant relationships, although ESO boys do. Significant ($\chi^2 = 4.115$, $p = 0.043$, Cc = 0.314) and inverse ($\Phi = -0.228$) relationships are observed between the degree of help at the school and the victimization of insulting behavior, feeling ridiculed, or recognizing how their partner or ex-partner has tried to make them believe they were worthless. Similarly, significant relationships with violent behavior are observed in couples exercised among ESO students and the degree of help in the school. Specifically, significant ($\chi^2 = 5.387$, $p = 0.020$, Cc = 0.359) and inverse ($\Phi = -0.263$) relationships in aggressor behavior related to insults, ridiculing, and making their partner believe that they are worthless, significant ($\chi^2 = 6.352$, $p = 0.012$, Cc = 0.387) and inverse ($\Phi = -0.285$) relationships in aggressor behaviors related to forcing their partner to do things they did not want to do, such as changing clothes and significant ($\chi^2 = 3.891$, $p = 0.049$, Cc = 0.309) and inverse ($\Phi = -0.225$) relationships in aggressor behaviors related to offenses or insults made via social networks or over the phone.

**Table 4.** Results of the Help Variable (CES) and the Observed Relationship with Violence in Adolescent Couples.

| | | | CES Help | | | |
|---|---|---|---|---|---|---|
| **Violence in Adolescent Couples** | | | **FPB Woman** | **ESO Woman** | **FPB Man** | **ESO Man** |
| Psychological violence | "My partner has insulted me, ridiculed me, or made me believe that I was worthless" 34.A. A | $\chi^2$ (*p*-value) N Cc $\Phi$ (*p*-value) $\Phi$ | - | - | - | 4115 0.043 79 0.314 −0.228 0.043 |
| Psychological violence | "I have insulted, ridiculed, or made my partner believe that he or she was worthless" 34.B. A | $\chi^2$ (*p*-value) N Cc $\Phi$ (*p*-value) $\Phi$ | - | - | - | 5387 0.020 78 0.359 −0.263 0.020 |
| Psychological violence | "I forced my partner to do things he or she did not want to do (change clothes, because those he or she were wearing did not seem good to me, etc.)" 34.B. C | $\chi^2$ (*p*-value) N Cc $\Phi$ (*p*-value) $\Phi$ | - | - | - | 6352 0.012 78 0.387 −0.285 0.012 |
| Psychological violence | "I have humiliated, insulted, or threatened my partner publicly or harassed him or her by the phone" 34.B. D | $\chi^2$ (*p*-value) N Cc $\Phi$ (*p*-value) $\Phi$ | - | - | - | 3891 0.049 77 0.309 −0.225 0.049 |
| SCALE: BEING AN AGGRESSOR OF VIOLENCE IN THE ADOLESCENT COUPLE | | $\chi^2$ (*p*-value) N Cc $\Phi$ (*p*-value) $\Phi$ | 4118 0.042 29 0.497 −0.377 0.042 | - | - | - |

More information about the questions used in Supplementary Materials.

### 4.3. Involvement Perceived in the School and Relationships Observed with Violence in Adolescent Couples (Victim and Victimizer)

Third, the data related to involvement (CES) is presented, referring to the degree to which students show interest in class activities and participate in discussions and how they enjoy the environment created by incorporating complementary tasks (Table 5).

**Table 5.** Results of the Involvement Variable (CES) and the Observed Relationship with Violence in Adolescent Couples.

| Violence in Adolescent Couples | | CES Involvement | | | |
|---|---|---|---|---|---|
| | | FPB Woman | ESO Woman | FPB Man | ESO Man |
| Psychological violence | "My partner has insulted me, ridiculed me, or made me believe that I was worthless" 34.A. A | $\chi^2$<br>(*p*-value)<br>N<br>Cc<br>Φ<br>(*p*-value) Φ | -<br><br><br><br> | -<br><br><br><br> | -<br><br><br><br> | 4104<br>0.043<br>80<br>0.311<br>−0.226<br>0.043 |
| Psychological violence | "I have insulted, ridiculed, or made my partner believe that he or she was worthless" 34.B. A | $\chi^2$<br>(*p*-value)<br>N<br>Cc<br>Φ<br>(*p*-value) Φ | 4342<br>0.037<br>30<br>0.502<br>−0.380<br>0.037 | -<br><br><br><br> | -<br><br><br><br> | -<br><br><br><br> |
| Psychological violence | "I tried to control (monitoring his or her cell phone, etc.) and isolate my partner from friends (forbidding him or her to see someone, etc.)" 34.B. B | $\chi^2$<br>(*p*-value)<br>N<br>Cc<br>Φ<br>(*p*-value) Φ | -<br><br><br><br> | 4558<br>0.033<br>93<br>0.305<br>−0.221<br>0.033 | -<br><br><br><br> | -<br><br><br><br> |

More information about the questions used in Supplementary Materials.

Female student body. In the case of FPB girls, the results show the existence of a significant ($\chi^2$ = 4.342, *p* = 0.037, Cc = 0.502) and inverse (Φ = −0.380) relationship between the degree of involvement in the school and being aggressors toward their partners, with insulting and ridiculing behavior and considering making them believe that they are worthless.

Between the ESO girls, a significant ($\chi^2$ = 4.558, *p* = 0.033, Cc = 0.305) and inverse (Φ = −0.221) relationship is observed between the degree of involvement in the school and recognizing oneself as an aggressor, controlling their partner or ex-partner's behavior and isolating them from friendships.

Male student body. No significant relationships are observed in relation to the degree of involvement in the center and violent behavior—from the victim or aggressor role—among FPB boys.

The ESO boys show significant ($\chi^2$ = 4.104, *p* = 0.043, Cc = 0.311) and inverse (Φ = −0.226) relationships between the degree of involvement in the center and feeling like victims of insults or ridicule by their partner and the feeling of having been made to believe that they were worthless.

After analyzing the three subscales that make up the CES (The Social Climate Scales: Classroom), we can conclude that the affiliation subscale shows the greatest number of significant relationships in both educational tracks, both among boys and girls. The data from the girls of both educational tracks are similar, finding a greater number of items in the ESO girls group that are inversely related to violence in adolescent couples. There is a trend of significant relationships between victimization behaviors, and limited, although existent (and relevant) among aggressors' behavior. It is therefore proven that a positive school climate, and affiliation in particular, influences less involvement among girls both as victims and aggressors in romantic relationships in psychological violence behaviors, showing a greater influence on the ESO female group.

Among the boys, "affiliation" is the subscale that shows the most significant relationship with violent behavior in romantic relationships. There are slight differences with respect to girls who only showed relationships with less involvement in psychological behavior. The ESO male student body shows significant relationships between an adequate degree of affiliation in the school and a lower involvement in behaviors of psychological violence from the perspective of victim and aggressor, but there are also relationships with less involvement as aggressors in physical and sexual violent behavior. The male FPB students also show inverse relationships between school affiliation and involvement as aggressors in psychological and sexual violence behaviors.

If we refer to the "perceived help" in the school subscale, the data only show significant results between the FPB girls and the ESO boys. With respect to FPB girls, the items that are inversely related

to violence in adolescent couples refer to the scale at which girls admit to being aggressors toward their partners. The ESO boys show significant and inverse relationships with involvement, both as victims and aggressors in psychological violence behaviors. It is therefore proven that a positive school climate, and in particular perceived help, influences FPB girls in less involvement as aggressors in romantic relationships and less involvement from ESO boys in psychological violence behavior, both from the victim role as well as from the aggressor role.

Finally, the involvement subscale again shows, and in the same way as the help subscale, significant and inverse results with the FPB girl group and FPB boy group. FPB girls show less involvement in behavior as aggressors related to psychological violence due to having an adequate degree of involvement in school, while ESO boys show less involvement in psychological violence behavior, in the role of victim, to perceive a better school involvement.

## 5. Discussion

In this research, we confirmed the relevance of schools at the stage of adolescence [24,32,33,40]. The educational involvement of the school climate is not only relevant but a clear line to work, because if we manage to implement safe, educational contexts and spaces where a proper and correct school climate is experienced [18,19] relative aggression and victimization in romantic relationships will decrease, working as a clear protective factor [49]. It is proven in this research that an adequate climate and a positive orientation toward school, a feeling of belonging and of group unity [16,24,33], social inclusion and educational support [16,26], cohesion, friendship with other classmates, involvement in the classroom [30,31], as well as participating in tutoring, meetings with the school community and changes in disciplinary procedures [33] are all measures that prevent risk behavior in adolescence. The results of this research show the relationship between a positive school climate and a diminished involvement in violent behavior among peers and violence in adolescent couples, as was already the case in previous studies [17].

Adolescent girls and boys are involved in violent behavior in romantic relationships in a similar way when we refer to psychological violence behaviors (control and isolation), which are the most common. We observed that in the most serious cases, in which besides psychological violence, violent behavior of another nature appears—sexual or physical—boys admit to being the aggressors. However, an adequate school climate, and an adequate degree of perception of affiliation, help, and involvement in the school may act as a protective factor against initiating these behaviors from early adolescence, to prevent their development, as well as work as a preventive factor in cases in which it takes place in middle or late adolescence, preventing them from developing and remaining in adulthood.

There is a stronger relationship between school climate and less involvement in violent behavior in adolescent couples among ESO students, possibly because it is a group that in principle is at lower risk or difficulty than FPB students, due to its inherent characteristics. Despite everything, in addition to the differences existing according to educational track, we can observe the existence of differences according to gender [14,15]. While the girls in both tracks show significant relationships between affiliation, help, and perceived involvement in the school and a lower involvement in psychological violence behavior, either as victims or aggressors, the boys, from both educational tracks, show not only existing relationship with the behavior of psychological violence but also their involvement in sexual and physical violence exercised.

Affiliation in the school, intimately related to a good relationship with the peer group, [17,21,26,27] shows important relationships with less involvement in violent behavior in romantic relationships, both from the role of victim and victimizer, especially among ESO students. In FPB students, the lower existence of significant relationships between affiliation and participation in violent relationships between peers can be explained by the accumulation of more risk factors. These factors cause adolescents to be immersed in violent situations, not working as a positive school climate, and affiliation in particular, enough influence to avoid involvement from the victim or victimizer role. Be that as it may, and although in the ESO group, there is a greater number of significant relationships,

adequate affiliation to the school is shown as a preventive variable, as well as help and involvement in the school [22–25]; mainly with behaviors of psychological violence in romantic relationships. Thus, feeling and being part of the group [34] will not determine non-involvement in violent behavior in romantic relationships but will influence it, minimizing its risk or impact. In addition, the perception of help and involvement of teachers [26,30,31] will act as a protective factor against involvement in violent behavior, especially among ESO students.

Therefore, it is important that schools can promote the resilient capacity of minors [4,37,50–52], encouraging the appearance and strengthening of protective factors, whether they are specific to minors or promoted by the educational center. Regarding the protective factors of the educational center, we must understand this space as a meeting place where teaching–learning takes place, not only from an academic point of view but also at a personal, social, and coexistence level, making the support and guidance provided by adult figures of reference very relevant. We must therefore ask ourselves about the skills that the professionals we place in schools possess to identify cases of violence, so as to tackle them in time, using the educational center as a key space in the prevention of various kinds of violence [23]. To this effect, as Pizzi [25] indicated, schools should be structured as welcoming, stimulating, and habitable places in which students can express and work on, with the required professional help, conflict experienced in the various areas of their lives, achieving development toward adulthood [18]. The main task, for both society and schools, will be based on transmitting the values and appropriate models of behavior, not falling into an empty space or lacking rules that can lead to situations of violence and situations based on anomie. In the words off Pérez Serrano ([53]- p. 8):

> *In this context, teachers are forced to redefine their roles: to stop being mere transmitters of knowledge in order to become guides and mediators between students and information. They will teach to select relevant content and assimilate them, to interrelate them, and put them into practice. This means that, more and more, the skills and competences that are needed outside school are prioritized. ( … ) Schools are called to conceptually and functionally redefine their being and doing, in order to give a response that is more in line with the needs of the society described. We must keep in mind that schools must prepare for the future, and before a new reality, a response should not be given from schools thought in another historical reality. In these circumstances, schools must consider new objectives, to which social education can offer new possibilities.*

Therefore, we support the idea [53] that, in addition to the teaching staff, other professionals in psychological and educational intervention who can attend to students should be appointed and new needs that are emerging, should be considered by taking in to account not only the purely educational-formative perspective, but also demand and new challenges, considering the diversity of social, personal, cultural, and religious situations that arise [19]. Social education should promote both coexistence and cooperation among students, creating full-fledged citizens who know how to resolve their conflicts and manage communication skills, both inside and outside school. If this perspective is implemented, it will promote an important social transformation, by forming citizens capable of reflecting autonomously and critically, who feel socially committed and develop and promote equity and social justice [19], and are therefore involved in less violent behavior, among other risk behaviors. Education is a vital tool for social transformation.

Educational centers, and the professionals who are involved in them, should support and promote educational responses that adolescents need in order to understand the conflict situations of this vital stage, among which we encounter violence in adolescent couples. If we educate our youth, we will change the future and society. Education is a priority intervention tool to reduce violence.

Regarding the limitations of this research, it should be mentioned that the main limitation observed is based on the need to expand the sample under study. Therefore, for future research, we would like to have a larger sample, to be able to delve into the socio-contextual influences, considering, in addition to the school's influence (and the climate found there), family influences and the impact of family climate in the development of violent behavior among peers, specifically in adolescent romantic

relationships. We also consider as a limitation the difficulty associated with addressing this issue, violence in adolescent couples, since they include questions of a very intimate nature, and therefore, the answers obtained about the experience of violence, and the types of violence both suffered and exercised (taking into account that they refer to the individual perception of it) may be biased.

**Supplementary Materials:** The following are available online at http://www.mdpi.com/2071-1050/12/11/4705/s1.

**Author Contributions:** Both M.R.-N. and R.S.G. have jointly produced the theoretical framework. Although the analyzes have been developed in greater depth by M.R.-N., R.S.G. is participating more actively in the discussion. Both authors have carried out the revision of format and edition. It has been a collaborative work. Both authors have read and agreed to the published version of the manuscript.

**Funding:** The Provincial Council of Bizkaia (DFB) in its Bizkailab program (5738-11), with the Bizkume project, and the research staff training grant (FPI-UD 2012-15), funded this research.

**Acknowledgments:** Thanks for the participation of all the adolescents and school centers that have participated in this investigation. As well as the support of the entities, that have financed the investigation.

**Conflicts of Interest:** The authors declare that they have no conflict of interest.

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
