# Peer review of "School Climate and Peer Victimization. Involvement, Affiliation and Help Perceived in School Centers as Protective Factors against Violent Behavior in Adolescent Couples"

_sustainability, doi:10.3390/su12114705_

Round 1

Reviewer 1 Report

Manuscript covers relevant and actual topic of violence in adolescents' close relationships and the role of school climate as protective factor in violence.  Introduction part is sound, review of previous research in the filed is presented. However, it is not clear what did authors expected; there are no hypotheses or research questions that would frame the introduction part and introduce readers to current research. 

Suggestions/comments:

1. In the Methodology section, procedure is poorly descibed, there are only some information in Ethical Issues section. I suggest that authors separate these sections (Procedure and Ethical Issues), or under Procedure section that they include ethical aspects of the research. 

2. There is no enough information on instruments used (what was the response scale, how did participants answered, descriptive statistics...). These information need to be added. 

3. Title of Table II should be more explicative

4. There is no explication for ESO or FPB abbreviations used in the text

5. Sample description regarding socio-demographic characteristics of sample should be in Methodology section (Participants) and not in the Results. Moreover, authors report data on characteristic that they did not use further in the analyses, therefore, it is not clear why report them in the first section of the Results. 

6. It is not clear why authors used only correlation to analyse their data. I think they could add more sophisticated statistical analyses on data they gathered. That way, they would have more sound foundation for conclusions they made. 

7. In the limitations part (at the end of discussion), authors only mentioned the sample size but there are other important issues that need to be adressed, for example relatively low realiability of CES subscales; the issue of socially desirable answers since the topis included very intimate questions on experiencing violence; bias in answering questions that refer to violence; shortcomings of statistical analyses they used etc. 

8. Throughout the manuscript, there are many too long sentences that make text difficult to read. Text needs certain refinement so that readers could follow authors ideas and conclusions. 

9. English editing is recommended

Author Response

Thanks for your comments:

1. We add an explanatory paragraph regarding the hypothesis presented.

2. We separated in the methodological section, as suggested by the procedure, and expanded the information related to Ethics.

3. We attach information regarding the methodological tools used (Annex 1)

4.We clarify the abbreviations "ESO" and "FPB".

5. We readjusted the first paragraph of results, incorporating it in the sample description, as suggested, also incorporating changes made by reviewer 2, in relation to how we refer to the immigrant population.

6. As indicated on the web, in the section on the suggestion to expand the analysis carried out, we maintain the initial proposal since we continue to consider it of interest, adjusting to the presentation of correlations, etc.

7.We extend the paragraph regarding limitations, taking into account the indications made by reviewer 1.

8. We attach, in relation to English, the official translation certificate made.

It is not clear why authors used only correlation to analyse their data. I think they could add more sophisticated statistical analyses on data they gathered. That way, they would have more sound foundation for conclusions they made.

There is no consistency between evaluators on this point (6) so the authors consider more appropriate to leave this point without modifications.

In relation to the rest of the suggestions, all have been taken into account.

Regarding the translation, indicate that it has been carried out by a translation company. We attach the certificate.

Reviewer 2 Report

Nowadays, given the worrying situation of gender violence among adolescents, research like this is very necessary. The social sciences must identify social problems and must also offer answers to them. For this reason, the authors are congratulated for shedding some light on the issue of gender violence among adolescents. In this sense, the focus of study of this work is very well chosen and well justified. Educational centers and the school climate, are essential for the prevention of situations of violence between equals. This article provides evidence of recent studies on the first pages. Also, this empirical study new and interesting data offered, that illuminate the problem a little more complex reality of gender violence among adolescents.

I think some small aspects could be improved before publication:

  1. In the description of the number of participants, percentages are offered between "native adolescents" and "adolescents of migrant origin". But then there is direct talk of "adolescent immigrants." Terminology should be unified. In any case, on this variable (migration) that are introduced in the description of the participants, then no specific data is provided in the results. Perhaps this specification in the participant description might not be necessary.
  2. The authors address the need for schools to promote resilience (line 405). Given the potential that this concept has developed in recent years, it would be interesting to attach some references and evidence from more recent works (perhaps from the last 5 years).
  3. Authors defend that professionals are not only transmitters of knowledge but that they help adolescents to empower themselves and be more co-responsible (line 411 and following). This line of discussion is of great interest and it would be desirable to include some specific recommendations and more current bibliographic references, about tasks that education professionals (in and out of school) can do to support and accompany adolescents in their conflicts with gender violence.

I reiterate my congratulations to the authors for a job with a strong commitment to subvert gender violence in our society.

Author Response

In relation to reviewer 2, comments that:

1. As indicated, we have unified the criteria and use of the concept "IMMIGRANT ORIGIN"

2. We have incorporated a current reference regarding resilience in the school environment.

3. We have not delved into bibliographic references related to the deepening of the intervention of professionals, since we consider that although it is an interesting line, it would be appropriate to delve into it in another article.

Reviewer 3 Report

The subject is interesting because it raises the attitudes of gender violence in adolescent relationships; as well as the role that the school has to play as an educational institution to promote gender relations under equal conditions. It places education at the center of all intervention, promoting education in equality and respect for individual liberties. It´s very interesting to analyze the data according to the level of study and the gender of the students; in order to promote actions to alleviate this problem in educational centers and in the social context of adolescents in Bizkaia and, by extension, in Spain in general.

Author Response

Thanks for your comments regarding our article.

Round 2

Reviewer 1 Report

I think the manuscript has been significantly improved and authors adequately used suggestions and comments I offered in the review. Still, I believe the authors could also rearrange analyses used for testing their hypothesis.